# Retrospective cohort study of anti-tumor necrosis factor agent use in a Veteran Population

Mark Bounthavong, Nermeen Madkour and Rashid Kazerooni

Veterans Affairs San Diego Healthcare System, San Diego, CA, USA

## ABSTRACT

**Introduction.** Anti-tumor necrosis factor (TNF) agents are effective for several immunologic conditions (rheumatoid arthritis (RA), Crohn's disease (CD), and psoriasis). The purpose of this study was to evaluate the efficacy and safety of anti-TNF agents via chart review.

**Methods.** Single-site, retrospective cohort study that evaluated the efficacy and safety of anti-TNF agents in veterans initiated between 2010 and 2011. Primary aim evaluated response at 12 months post-index date. Secondary aims evaluated initial response prior to 12 months post-index date and infection events.

**Results.** A majority of patients were prescribed anti-TNF agents for CD (27%) and RA (24%). Patients were initiated on etanercept (41%), adalimumab (40%), and infliximab (18%) between 2010 and 2011. No differences in patient demographics were reported. Response rates were high overall. Sixty-five percent of etanercept patients, 82% of adalimumab patients, and 59% of infliximab patients were either partial or full responders, respectively. Approximately 16%, 11%, and 12% of etanercept, adalimumab, and infliximab were non-responders, respectively. Infections between the groups were non-significant. Etanercept and adalimumab patients had higher but non-significant odds of being a responder relative to infliximab.

**Conclusions.** Most patients initiated with anti-TNF agent were responders at 12 months follow-up for all indications in a veteran population.

## INTRODUCTION

In the past two decades, biologic therapies have reshaped how clinicians approached chronic disease management (*Agarwal, 2011a*; *Agarwal, 2011b*; *Ford et al., 2011*; *Lichtenstein, Hanauer & Sandborn, 2009*; *Mayberry et al., 2013*; *Singh & Cameron, 2012*). Immunologic disorders such as rheumatoid arthritis (RA) and Crohn's disease (CD) have traditionally relied on oral pharmacotherapy for treatment of acute symptoms, management, and remission. However, oral therapies were unable to provide long-term control and disease progression resulting in relapse and hospital admission/surgery. Biologic agents, such as monoclonal antibodies, target the host's immune system to attenuate the self-destructive immune response, which is the cause of RA and CD. Clinical

Corresponding author
Mark Bounthavong,
mbounthavong@outlook.com

efficacy with biologics has been reported in RA and CD as well as a reduction in hospital admission/surgery (*Bodger, 2002*; *Lundkvist, Kastäng & Kobelt, 2008*). More importantly, biologic therapy has improved the quality of life for patients suffering with these chronic diseases (*Feagan et al., 2009*; *Staples et al., 2011*).

Monoclonal antibodies, in particular, the anti-tumor necrosis factor (TNF) agents, have demonstrated significant reductions in disease symptoms, progression, and improvement in patient quality of life (*Feagan et al., 2009*; *Ford et al., 2011*; *Lundkvist, Kastäng & Kobelt, 2008*; *Nixon, Bansback & Brennan, 2007*; *Ordás, Feagan & Sandborn, 2011*). In several studies, anti-TNF agents have increased the proportion of patients who experience remission; thereby, controlling the disease and limiting permanent damage. In some studies, remission duration has been reported for several years (*Ancuţa et al., 2009*; *Emery et al., 2010*; *Van der Heijde et al., 2006*).

RA is a systemic autoimmune disorder which is characterized by inflammation of the synovial joints (*Segal, Rhodus & Patel, 2008*). RA affects about 0.5% to 1.0% of the US population with a prevalence of 1.3 million (*Gabriel & Michaud, 2009*; *Helmick et al., 2008*). The health burden of RA in the US was estimated to be 98 Disability Adjusted Life Years (DALYS) lost per 100,000 population; and 1 RA-related death per 100,000 population (*Lundkvist, Kastäng & Kobelt, 2008*). In the VA, there were a total of 1,694 RA-related mortalities from 1999 to 2004 (*Lee et al., 2007*). The age-adjusted 5-year RA-related mortality rate among patients with a single condition relative to no other condition was 6.05 (95% confidence interval [CI] [4.90, 7.20]) (*Lee et al., 2007*). The average annual costs of RA per person in the US was $12,558 (adjusted for 2006 $US) (*Lundkvist, Kastäng & Kobelt, 2008*).

The goal of therapy for patients with RA is to control and reduce the rate of degeneration of the joints due to immunologic destruction by the host's immune system (*Agarwal, 2011a*). In addition, quality of life and increased productivity are important milestones for treatment. Anti-TNF agents have been reported to reduce the rate of radiographic progression and improve short-term inflammatory symptoms (*Bathon et al., 2000*; *Breedveld et al., 2006*; *Choy et al., 2012*; *Emery et al., 2009*; *Keystone et al., 2008*; *Keystone et al., 2009*; *Keystone et al., 2004*; *Klareskog et al., 2004*; *Maini et al., 1999*; *Moreland et al., 1999*; *St Clair et al., 2004*; *Van de Putte et al., 2004*; *Weinblatt et al., 2003*; *Weinblatt et al., 1999*). Consequently, improvement in clinical outcomes has resulted in improved quality of life for RA patients. To date, there are five FDA-approved anti-TNF agents for RA: adalimumab (Humira®), certolizumab pegol (Cimzia®), etanercept (Enbrel®), golimumab (Simponi®), and infliximab (Remicade®) (*Agarwal, 2011b*).

Crohn's disease is a chronic inflammation of the gastrointestinal tract that is characterized by abdominal pain, diarrhea, gastrointestinal bleeding, bowel perforations, and fistulas (*Baumgart & Sandborn, 2012*). The incidence of Crohn's disease in the United States (US) was 7.9 cases per 100,000 population (1990–2000); and the adjusted prevalence was 174 per 100,000 population (2001) (*Loftus et al., 2007*; *Loftus, Schoenfeld & Sandborn, 2002*). In 2009, the average annual age- and gender-adjusted incidence rate of CD among veterans was 33 per 100,000 population (range: 27–40) (*Hou et al., 2013*). The age- and

gender-adjusted point prevalence of CD among veterans was 287 per 100,000 population (*Hou et al., 2013*). Prior to the widespread use of anti-TNF agents, the average annual cost per patient in the US was estimated to be $19,237 (adjusted for 2012 $US) with surgery responsible for a majority of direct costs (55.8%) (*Bodger, 2002*). However, after the widespread use of anti-TNF agents, the average annual cost per patient with CD was $13,699 per year (adjusted for 2012 $US) (*Kappelman et al., 2008*).

Biologic therapies, such as anti-TNF agents, for Crohn's disease have provided clinically meaningful improvement in patient reported outcomes while maintaining remission (*Ford et al., 2011*; *Hanauer et al., 2006*; *Louis et al., 2013*; *Sandborn et al., 2007a*; *Sandborn et al., 2007b*). As a result, the increased utilization of anti-TNF therapy has shifted costs from hospitalizations and surgeries to medications. *Van der Valk et al. (2012)* reported that medication costs were responsible for 70.9% of total direct costs compared to hospitalizations- (19.4%) and surgery-related costs (0.6%) in the Netherlands (*Van der Valk et al., 2012*). *Loomes et al. (2011)* reported that total direct costs increased from $3,930 to $25,346 (difference of $21,416, $P < 0.005$) after the introduction of infliximab therapy (adjusted for 2010 $CAN) (*Loomes et al., 2011*). Currently, there are three anti-TNF agents FDA-approved for the treatment and management of CD: adalimumab, certolizumab pegol, and infliximab (*FDA Office of the Commissioner, 2008*, *National Digestive Diseases Information Clearinghouse (NDDIC)*).

The Department of Veterans Affairs has a national formulary that is shared with all the VA medical centers around US and its territories. However, none of the anti-TNF agents are listed on the VA National Formulary (VANF) as of August 2013. This is important because the burden of disease in the VA is significant. There have been no reports that currently investigated the efficacy and safety of anti-TNF agents in the veteran population for all indications.

The purpose of this study was to evaluate the efficacy and safety of anti-TNF agent use in the Veterans Affairs San Diego Healthcare System (VASDHS) who initiated therapy in 2010 and 2011 for all prescribed indications. Particular attention was focused on RA and CD due to early approvals in these therapeutic areas.

## METHODS

This was a single-site, retrospective cohort study that evaluated the efficacy and safety of anti-TNF agents in a veteran population who initiated treatment between 2010 and 2011 and followed-up for 12 months. The study site was at VASDHS, a 296-bed medical facility in the San Diego County, California with a regional patient membership of approximately 232,000 veterans. VASDHS is part of the Veterans Health Administration (VHA), an integrated healthcare system in the US.

Patients were eligible for inclusion if they were 18 years old or greater and initiated on an anti-TNF agent at VASDHS between 2010 and 2011. The index date was determined to be the first fill-date of the anti-TNF agent at VASDHS.

Clinical efficacy was categorized as responder, partial responder, and non-responder which were determined from chart notes as defined by the provider. Responders were

defined as any documented report of improvement from baseline based on resolution of symptoms and clinical assessment by the provider. Partial responders were defined as any documented report of partial improvement from baseline based on attenuated but continued symptoms and clinical assessment by the provider. Non-responders were defined as any documented report of no improvement from baseline based on continued or worsening of symptoms and clinical assessment by the provider. Two reviewers independently performed the chart reviews (MB and NM) and any disagreements on clinical response were resolved through group discussion.

Primary indication for the anti-TNF agent was determined through the submission of non-formulary (or prior authorization) consults which were reviewed by the VASDHS pharmacy service pharmacoeconomics/formulary group. Anti-TNF agents are listed as non-formulary in the VHA; therefore, requests for these agents in VASDHS require a submission of a non-formulary consult. Providers were required to list the primary indication for anti-TNF agent use. If more than one indication was listed, then the primary indication was categorized according to the specialty field of the submitting provider. For example, a rheumatology provider who submitted a non-formulary consult for both arthritis and psoriasis will have the indication categorized for RA.

Primary aim evaluated response at 12 months post-index date. A majority of clinical trails evaluated response at 12 months; therefore, we also followed this convention. Secondary aims evaluated initial response to anti-TNF agents prior to the 12 months post-index date, alternative strategy after failure to respond or development of an adverse drug event to the initial anti-TNF agent, and infection events. Reporting was further stratified into the top three indications: RA, CD, and psoriasis. Infection events included any infection that occurred after the index date up to 12 months post-index date.

This study received appropriate approvals from the UCSD/VASDHS Institutional Review Board and the Research and Development Committee (Protocol #: H120150).

## Statistical analysis

Normality testing was performed using Shapiro–Wilk's test for continuous data. Descriptive analyses for continuous data were presented as mean, standard deviation, and median. Discrete data were presented as frequency and percentage. One-way analysis of variance and Kruskal–Wallis tests were performed for continuous data where appropriate. Pearson's chi-squared and Fisher's exact tests were performed for discrete data.

Logistic regression was performed to evaluate the association between anti-TNF agents and response controlling for potential confounders. The outcome variable was transformed into a binary variable in order to perform the logistic regression. Responders and partial responders were collapsed into "Responders". Non-responder and patients who experienced an adverse drug event were categorized as "Non-responders". Model fit was assessed using Hosmer–Lemeshow test. Statistical significance was defined as $P < 0.05$, two-tailed. All analyses were performed using IBM SPSS Statistics for Windows, Version 20.0 (IBM Corp., Armonk, NY).

## RESULTS

### Baseline

A total of 92 patients met the inclusion criteria. Table 1 summarizes the demographic variables of the cohort. The average patient was 50 (SD, 16.2) years old, male ($N = 77$, 84%), non-Hispanic ($N = 78$, 85%), and white ($N = 68$, 74%). CD was the most common indication for an anti-TNF agent ($N = 25$, 27%) followed by RA ($N = 22$, 24%), psoriasis ($N = 19$, 21%), psoriatic arthritis ($N = 13$, 14%), other conditions ($N = 8$, 9%), and ankylosing spondylitis ($N = 5$, 5%). The most common comorbid conditions were hypertension ($N = 39$, 42%), dyslipidemia ($N = 36$, 39%), gastrointestinal conditions excluding CD ($N = 24$, 26%), cardiovascular disease ($N = 11$, 12%), and diabetes ($N = 11$, 12%). Several patients were on prednisone ($N = 18$, 20%) or methotrexate ($N = 15$, 16%) at baseline. Less than half of the study patients had previous experience with an anti-TNF agent ($N = 42$, 46%), most commonly adalimumab ($N = 22$) followed by etanercept ($N = 11$) and infliximab ($N = 9$).

A majority of patients were started on adalimumab ($N = 38$) and etanercept ($N = 37$) followed by infliximab ($N = 17$) between 2010 and 2011 at VASDHS (Table 2). There were no differences in age ($P = 0.141$), gender ($P = 0.480$), ethnicity ($P = 0.132$), and race ($P = 0.726$) between the three anti-TNF agents. No difference in primary diagnosis for anti-TNF agent use was reported with RA ($P = 0.119$), psoriatic arthritis ($P = 0.167$), ankylosing spondylitis ($P = 0.474$), and other conditions ($P = 0.157$) between the three anti-TNF agents. Infliximab and adalimumab were often used in CD compared to etanercept ($P < 0.0001$). Conversely, a majority of patients received adalimumab to treat psoriasis relative to the other agents ($P < 0.0001$). There were no statistically significant difference in comorbidities between the three anti-TNF agents except for hypertension ($P = 0.023$), other gastrointestinal conditions other than CD ($P = 0.016$), and hypothyroidism ($P = 0.020$). A majority of patients had tuberculosis screening ($N = 83$, 90%) and hepatitis B screening ($N = 73$, 79%) performed at baseline.

At baseline, methotrexate was only reported by patients who started on etanercept ($N = 8$) and adalimumab ($N = 7$). A small number of prednisone prescriptions were written at baseline during initiation of etanercept ($N = 6$), adalimumab ($N = 8$), and infliximab ($N = 4$). Among patients who started on etanercept at the VASDHS, six had previous experience with it. Similarly, among patients who were initiated on adalimumab and infliximab at VASDHS, eleven and two patients had a previous history with those agents, respectively.

### Clinical response

The average time to first follow-up visit was 86 (SD, 120) days. At the initial follow-up, 73 (83%) patients responded (responder and partial responder) to therapy (Table 3). At 12 months follow-up, a majority of patients responded (responder and partial responder) to therapy ($N = 65$, 71%). After 12 months of follow-up, there were 15 unique cases (16%) of infections that did not require hospital admissions, and three adverse drug events were reported which resulted in discontinuation of anti-TNF agent therapy. Two of the

**Table 1 Demographics of entire cohort started on anti-tumor necrosis factor (TNF) agents, 2010–2011.**

| N | 92 | |
|---|---|---|
| **Variable** | **Mean** | **SD** |
| Age (years) | 49.97 | 16.23 |
| Body mass index (kg/m$^2$) | 28.96 | 5.49 |
| Aspartate aminotransferase (mg/dL) | 24.88 | 20.35 |
| Alanine aminotransferase (mg/dL) | 28.13 | 25.71 |
| | **Number** | **Percent** |
| Gender | | |
|     Male | 77 | 84% |
|     Female | 15 | 16% |
| Ethnicity | | |
|     Hispanic | 13 | 14% |
|     Non-Hispanic | 78 | 85% |
|     Unknown | 1 | 1% |
| Race | | |
|     White | 68 | 74% |
|     Black | 11 | 12% |
|     Asian | 3 | 3% |
|     Native American/Pacific Islander | 2 | 2% |
|     American Indian/Alaskan Native | 1 | 1% |
|     Unknown | 5 | 5% |
|     Declined | 2 | 2% |
| Primary Diagnosis | | |
|     Rheumatoid arthritis | 23 | 25% |
|     Crohn's disease | 24 | 26% |
|     Psoriasis | 19 | 21% |
|     Psoriatic arthritis | 13 | 14% |
|     Other[*] | 7 | 8% |
|     Ankylosing spondylitis | 5 | 5% |
| Comorbid conditions | | |
|     Diabetes | 11 | 12% |
|     Hypertension | 39 | 42% |
|     Arrhythmia | 3 | 3% |
|     Heart failure | 3 | 3% |
|     Malignancy | 7 | 8% |
|     Chronic lung disease | 5 | 5% |
|     Cardiovascular disease | 11 | 12% |
|     Hepatic disease | 3 | 3% |
|     Renal | 5 | 5% |
|     Gout | 5 | 5% |
|     Hepatitis C | 4 | 4% |
|     Dyslipidemia | 36 | 39% |

Table 1 (*continued*)

|  | Number | Percent |
|---|---|---|
| History of myocardial infarction | 2 | 2% |
| Gastrointestinal (other than Crohn's disease) | 24 | 26% |
| Hypothyroidism | 5 | 5% |
| **Baseline DMARDS** | | |
| Methotrexate | 15 | 16% |
| Prednisone | 18 | 20% |
| Sulfasalazine | 9 | 10% |
| Plaqguenil | 4 | 4% |
| **Previous anti-TNF agent** | | |
| Yes | 42 | 46% |
| No | 50 | 54% |
| **Anti-TNF agent history** | | |
| Adalimumab history | 22 | 24% |
| Etanercept history | 11 | 12% |
| Infliximab history | 9 | 10% |
| **Anti-TNF agent history origin** | | |
| Community provider | 21 | 23% |
| Another VA facility | 5 | 5% |
| Department of Defense | 4 | 4% |
| Veterans Affairs San Diego Healthcare System | 12 | 13% |
| **Rheumatoid factor result at baseline** | | |
| Positive | 11 | 12% |
| Negative | 14 | 15% |
| **Tuberculosis test performed** | | |
| Yes | 83 | 90% |
| No | 9 | 10% |
| **Tuberculosis result** | | |
| Positive | 3 | 3% |
| Negative | 79 | 86% |
| **Hepatitis test performed** | | |
| Yes | 73 | 79% |
| No | 19 | 21% |
| Hepatitis B surface antigen (+) | 27 | 29% |
| Hepatitis B surface antibody (+) | 1 | 1% |
| Hepatitis C antibody (+) | 7 | 8% |

**Notes.**

[*] "Other" includes ulcerative colitis ($N = 5$), uveitis ($N = 1$), and spondylarthropathy ($N = 1$).

drug events that resulted in discontinuation were infection-related (abscess and surgical wound); the other was for myelosplastic syndrome.

At 12 months follow up, there was no significant differences in responses between anti-TNF agents ($P = 0.904$). In patients initiated on etanercept, 18 (49%) were responders, 6 (16%) were partial responders, 6 (16%) were non-responders, and 2 (5%) had an adverse drug event (myelospastic syndrome and surgical wound infection) at 12 months (Fig. 1). In patients initiated on adalimumab, 23 (61%) were responders, 8 (21%)
**Table 2 Demographics of patients initiated on etanercept, adalimumab, and infliximab, 2010–2011.**

| | Etanercept | | | Adalimumab | | | Infliximab | | | P-value |
|---|---|---|---|---|---|---|---|---|---|---|
| N | 37 | | | 38 | | | 17 | | | |
| Variable | Mean | SD | Median | Mean | SD | Median | Mean | SD | Median | |
| Age (years) | 52.92 | 15.15 | 56.0 | 49.47 | 15.97 | 55.0 | 44.60 | 17.69 | 41.0 | 0.141 |
| Body mass index (kg/m$^2$) | 30.44 | 6.02 | 30.2 | 28.30 | 4.92 | 28.5 | 27.02 | 4.79 | 26.7 | 0.108 |
| Aspartate aminotransferase (mg/dL) | 28.57 | 28.06 | 22.0 | 24.00 | 13.43 | 21.0 | 18.31 | 6.02 | 18.0 | 0.122 |
| Alanine aminotransferase (mg/dL) | 33.59 | 34.75 | 24.0 | 27.92 | 17.59 | 230 | 16.00 | 5.29 | 15.0 | 0.004 |

| | Etanercept | | Adalimumab | | Infliximab | | Chi-square | df | P-value |
|---|---|---|---|---|---|---|---|---|---|
| | Number | Percent | Number | Percent | Number | Percent | | | |
| Gender | | | | | | | | | |
| Male | 33 | 89% | 30 | 79% | 14 | 82% | 1.469 | 2 | 0.480 |
| Female | 4 | 11% | 8 | 21% | 3 | 18% | | | |
| Ethnicity | | | | | | | | | |
| Hispanic | 4 | 11% | 8 | 21% | 1 | 6% | 7.069 | 4 | 0.132 |
| Non-Hispanic | 33 | 89% | 30 | 79% | 15 | 88% | | | |
| Unknown | 0 | 0% | 0 | 0% | 1 | 6% | | | |
| Race | | | | | | | | | |
| White | 29 | 78% | 25 | 66% | 14 | 82% | 8.724 | 12 | 0.726 |
| Black | 5 | 14% | 5 | 13% | 1 | 6% | | | |
| Asian | 1 | 3% | 1 | 3% | 1 | 6% | | | |
| Native American/ Pacific Islander | 1 | 3% | 1 | 3% | 0 | 0% | | | |
| American Indian/ Alaskan Native | 0 | 0% | 1 | 3% | 0 | 0% | | | |
| Unknown | 1 | 3% | 4 | 11% | 0 | 0% | | | |
| Declined | 0 | 0% | 1 | 3% | 1 | 6% | | | |

Table 2 (*continued*)

| | Etanercept | | Adalimumab | | Infliximab | | Chi-square | df | P-value |
|---|---|---|---|---|---|---|---|---|---|
| | Number | Percent | Number | Percent | Number | Percent | | | |
| **Primary diagnosis** | | | | | | | | | |
| Rheumatoid arthritis | 10 | 27% | 12 | 32% | 1 | 6% | 4.272 | 2 | 0.119 |
| Crohn's disease | 0 | 0% | 12 | 32% | 12 | 71% | 31.113 | 2 | <0.0001 |
| Psoriatic arthritis | 7 | 19% | 6 | 16% | 0 | 0% | 3.583 | 2 | 0.167 |
| Ankylosing spondylitis | 3 | 8% | 2 | 5% | 0 | 0% | 1.494 | 2 | 0.474 |
| Psoriasis | 16 | 43% | 3 | 8% | 0 | 0% | 19.722 | 2 | <0.0001 |
| Other* | 1 | 3% | 3 | 8% | 3 | 18% | 3.708 | 2 | 0.157 |
| **Comorbid conditions** | | | | | | | | | |
| Diabetes | 7 | 19% | 4 | 11% | 0 | 0% | 4.086 | 2 | 0.130 |
| Hypertension | 21 | 57% | 15 | 39% | 3 | 18% | 7.521 | 2 | 0.023 |
| Arrhythmia | 1 | 3% | 2 | 5% | 0 | 0% | 1.093 | 2 | 0.579 |
| Heart failure | 0 | 0% | 3 | 8% | 0 | 0% | 4.407 | 2 | 0.110 |
| Malignancy | 2 | 5% | 5 | 13% | 0 | 0% | 3.32 | 2 | 0.190 |
| Chronic lung disease | 1 | 3% | 3 | 8% | 1 | 6% | 0.991 | 2 | 0.609 |
| Cardiovascular disease | 5 | 14% | 6 | 16% | 0 | 0% | 2.924 | 2 | 0.232 |
| Hepatic disease | 3 | 8% | 0 | 0% | 0 | 0% | 4.61 | 2 | 0.100 |
| Renal disease | 2 | 5% | 3 | 8% | 0 | 0% | 1.425 | 2 | 0.491 |
| Gout | 3 | 8% | 2 | 5% | 0 | 0% | 1.494 | 2 | 0.474 |
| Hepatitis C | 2 | 5% | 1 | 3% | 1 | 6% | 0.465 | 2 | 0.793 |
| Dyslipidemia | 16 | 43% | 17 | 45% | 3 | 18% | 4.058 | 2 | 0.131 |
| History of MI | 1 | 3% | 1 | 3% | 0 | 0% | 0.464 | 2 | 0.793 |
| GI (other than CD) | 10 | 27% | 14 | 37% | 0 | 0% | 8.297 | 2 | 0.016 |
| Hypothyroidism | 5 | 14% | 0 | 0% | 0 | 0% | 7.86 | 2 | 0.020 |
| **Baseline DMARDS** | | | | | | | | | |
| Methotrexate | 8 | 22% | 7 | 18% | 0 | 0% | 4.203 | 2 | 0.122 |
| Prednisone | 6 | 16% | 8 | 21% | 4 | 24% | 0.487 | 2 | 0.784 |
| Sulfasalazine | 5 | 14% | 2 | 5% | 2 | 12% | 4.013 | 2 | 0.134 |
| Plaguenil | 2 | 5% | 2 | 5% | 0 | 0% | 0.949 | 2 | 0.622 |

Table 2 (*continued*)

| | Etanercept | | Adalimumab | | Infliximab | | Chi-square | df | *P*-value |
|---|---|---|---|---|---|---|---|---|---|
| | Number | Percent | Number | Percent | Number | Percent | | | |
| **Previous TNF agent** | | | | | | | | | |
| Yes | 13 | 35% | 20 | 53% | 9 | 53% | 2.76 | 2 | 0.252 |
| No | 24 | 65% | 18 | 47% | 8 | 47% | | | |
| **Origin** | | | | | | | | | |
| Community provider | 8 | 22% | 9 | 24% | 4 | 24% | 3.317 | 6 | 0.768 |
| Another VA facility | 1 | 3% | 2 | 5% | 2 | 12% | | | |
| DoD | 1 | 3% | 3 | 8% | 0 | 0% | | | |
| VASDHS | 3 | 8% | 6 | 16% | 3 | 18% | | | |
| **RF result at baseline** | | | | | | | | | |
| Positive | 7 | 19% | 3 | 8% | 1 | 6% | 2.279 | 2 | 0.320 |
| Negative | 5 | 14% | 8 | 21% | 1 | 6% | | | |
| **TB test performed** | | | | | | | | | |
| Yes | 35 | 95% | 33 | 87% | 15 | 88% | 1.369 | 2 | 0.504 |
| No | 2 | 5% | 5 | 13% | 2 | 12% | | | |
| **TB result** | | | | | | | | | |
| Positive | 1 | 3% | 1 | 3% | 1 | 6% | 0.475 | 2 | 0.789 |
| Negative | 34 | 92% | 31 | 82% | 14 | 82% | | | |
| **Hepatitis test performed** | | | | | | | | | |
| Yes | 31 | 84% | 31 | 82% | 11 | 65% | 2.784 | 2 | 0.249 |
| No | 6 | 16% | 7 | 18% | 6 | 35% | | | |
| HBsAg (+) | 9 | 24% | 12 | 32% | 6 | 35% | 0.449 | 2 | 0.799 |
| HBsAb (+) | 1 | 3% | 0 | 0% | 0 | 0% | 1.33 | 2 | 0.514 |
| HCAb (+) | 5 | 14% | 2 | 5% | 0 | 0% | 2.457 | 2 | 0.293 |

**Notes.**

GI, gastrointestinal; CD, Crohn's disease; MI, myocardial infarction; RF, rheumatoid factor; TB, tuberculosis; HBsAg, hepatitis B surface antigen; HBsAb, hepatitis B surface antibody; HCAb, hepatitis C antibody.

* "Other" includes ulcerative colitis ($N = 5$), uveitis ($N = 1$), and spondylarthropathy ($N = 1$).

**Table 3** Outcomes at the first follow-up visit and at 12 months for patients started on etanercept, adalimumab, and infliximab at the VASDHS, 2010–2011.

| | All groups | | Etanercept | | Adalimumab | | Infliximab | | Chi-square | df | P-value |
|---|---|---|---|---|---|---|---|---|---|---|---|
| | Number | % | Number | % | Number | % | Number | % | | | |
| **Initial outcome at first follow-up visit** | | | | | | | | | | | |
| Responder | 65 | 71% | 23 | 62% | 27 | 71% | 15 | 88% | 7.764 | 4 | 0.101 |
| Partial | 11 | 12% | 7 | 19% | 4 | 11% | 0 | 0% | | | |
| Non-responder | 10 | 11% | 4 | 11% | 6 | 16% | 0 | 0% | | | |
| **Outcome at 12 months** | | | | | | | | | | | |
| Responder | 49 | 53% | 18 | 49% | 23 | 61% | 8 | 47% | 2.169 | 6 | 0.904 |
| Partial Responder | 16 | 17% | 6 | 16% | 8 | 21% | 2 | 12% | | | |
| Non-responder | 12 | 13% | 6 | 16% | 4 | 11% | 2 | 12% | | | |
| ADR | 3 | 3% | 2 | 5% | 1 | 3% | 0 | 0% | | | |
| **Infections after anti-TNF agent initiation** | | | | | | | | | | | |
| Yes | 15 | 16% | 5 | 14% | 10 | 26% | 0 | 0% | 6.314 | 2 | 0.043 |
| No | 77 | 84% | 32 | 86% | 28 | 74% | 17 | 100% | | | |

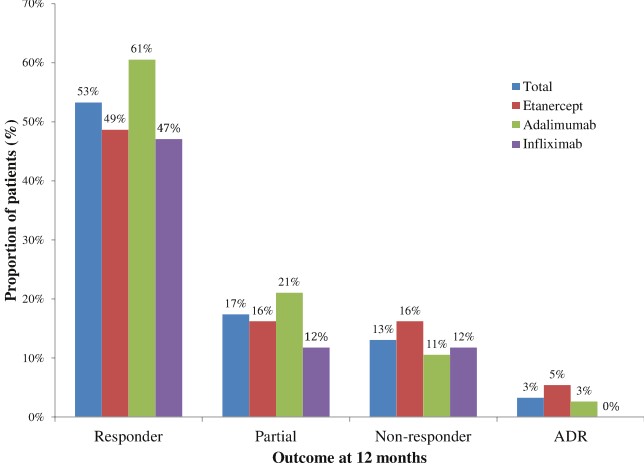

**Figure 1** Outcome with tumor necrosis factor use at 12 months, 2010–2011.

were partial responders, 4 (11%) were non-responders, and 1 (3%) had an adverse drug event (abscess) at 12 months. In patients initiated on infliximab, 8 (47%) were responders, 2 (12%) were partial responders, 2 (12%) were non-responders, and 0 had an adverse drug event at 12 months. There were missing data for 5, 2 and 5 patients in the etanercept, adalimumab, and infliximab groups, respectively. These missing data were considered missing completely at random; therefore complete-case analysis was appropriate (*Little & Rubin, 2002*).

Responders were stratified by RA, CD, and psoriasis for each anti-TNF agent (Fig. 2). In RA, 91% of patients receiving adalimumab were responders compared to 78% with etanercept. In CD, 89% of patients receiving infliximab were responders compared to 73%

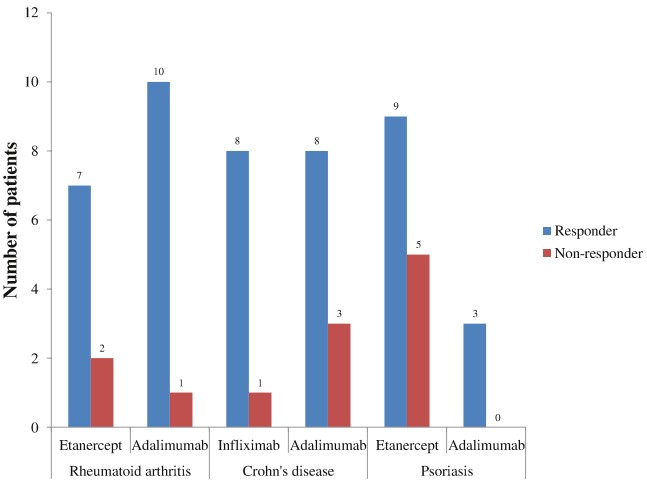

**Figure 2  Outcomes of different anti-TNF agents stratified by the top three disease states, 2010–2011.**

**Table 4  Odds of responder relative to infliximab.**

| Variable | B | SE | OR | 95% CI |
|---|---|---|---|---|
| **Crude analysis**[**] | | | | |
| Etanercept | −0.511[*] | 0.876 | 0.60 | 0.108, 3.338 |
| Adalimumab | 0.215[*] | 0.912 | 1.24 | 0.207, 7.412 |
| **Odds of responder adjusted for age, gender, and TNF history relative to infliximab**[***] | | | | |
| Etanercept | −0.090[*] | 0.979 | 0.91 | 0.134, 6.225 |
| Adalimumab | 0.613[*] | 1.000 | 1.85 | 0.260, 13.098 |
| Age, years | −0.064 | 0.024 | 0.94 | 0.895, 0.983 |
| Male | 0.351 | 0.919 | 1.42 | 0.234, 8.600 |
| TNF history | −0.161 | 0.646 | 0.85 | 0.240, 3.023 |

**Notes.**

[*] Referent is Infliximab.

[**] Hosmer–Lemeshow test, Chi-square $<0.0001$, $df = 1$, $P = 1.000$.

[***] Hosmer–Lemeshow test, Chi-square $= 9.670$, $df = 8$, $P = 0.289$.

with adalimumab. In psoriasis, 100% of patients receiving adalimumab were responders compared to 64% receiving etanercept.

Infections were reported for 5 (14%), 10 (26%), and 0 (0%) patients in the etanercept, adalimumab, and infliximab groups, respectively. This difference in infection rates between all three anti-TNF agents was statistically significant ($P = 0.043$).

Unadjusted odds of being a responder were 0.60 (95% CI [0.11, 3.34]) and 1.24 (95% CI [0.21, 7.41]) for patients initiated on etanercept and adalimumab relative to infliximab, respectively (Table 4). Controlling for age, gender, and previous history of anti-TNF agent use, the odds of being a responder was 0.91 (95% CI [0.13, 6.23]) and 1.85 (95% CI [0.26, 13.10]) for patients initiated on etanercept and adalimumab relative to infliximab, respectively.

## DISCUSSION

At VASDHS, patients initiated on an anti-TNF agent had a high proportion classified as responder (responder and partial responder) after 12 months of therapy. Reports from several clinical studies support this observation. *Weinblatt et al. (2003)* reported that 67% of patients randomized into adalimumab 40 mg every 2 weeks plus methotrexate for RA achieved American College of Rheumatology 20% (ACR20) at 24-week follow-up (*Weinblatt et al., 2003*). *Kameda et al. (2010)* reported that 90% of patients randomized into etanercept 25 mg twice weekly for RA achieved ACR20 at 24-week follow-up (*Kameda et al., 2010*). *Colombel et al. (2010)* investigated the efficacy of infliximab 5 mg per kg plus azathioprine in CD over a 30 week period and reported a remission rate of 57% (*Colombel et al., 2010*). *Sandborn et al. (2007b)* evaluated the long-term effectiveness of adalimumab 40 mg weekly and 40 mg every other week over 56 weeks in moderate-to-severe CD (*Sandborn et al., 2007b*). Remission was maintained in 83% and 79% of patients taking adalimumab 40 mg weekly and adalimumab 40 mg every other week, respectively (*Sandborn et al., 2007b*).

*Ng, Chu & Khan (2013)* performed a retrospective cohort study of biologic utilization for RA in the VA population from 1999 to 2009 (*Ng, Chu & Khan, 2013*). Biologics used as the first DMARD increased from 3% in 1999–2001 to 6.7% in 2006–2007 ($P < 0.001$) (*Ng, Chu & Khan, 2013*). However, the proportion of patients who had a biologic dispensed for RA was stable over the years ranging from 18.6% to 26.7% (*Ng, Chu & Khan, 2013*). We reported that 17% of patients who initiated etanercept previously had been on an anti-TNF agent; and 90% of patients who were initiated on adalimumab at VASDHS had previous experience with an anti-TNF agent. We adjusted for this in the logistic regression model and found that there was no significant confounding with previous history of anti-TNF agent use on the exposure-outcome relationship. A concern with previous anti-TNF agent use is confounding by indication where patients are inherently different due to severity of their disease which results in residual confounding (*Salas, Hofman & Stricker, 1999*). Future studies will need to address whether previous history of anti-TNF therapy has an impact on outcomes at 12 months follow up.

Utilization of anti-TNF agents in the CD veteran population has not been previously performed. However, an evaluation of hospitalization associated with CD in veterans was performed by Sonnenberg and colleagues (*Sonnenberg, Richardson & Abraham, 2009*). From 1975 to 2006, the total number of hospitalizations associated with CD among veterans was 54,271 with the highest proportion in the 54–64 year age group ($N = 22$, 551) (*Sonnenberg, Richardson & Abraham, 2009*). The incidence rate for hospitalization was 11.63 per 1 million population (*Sonnenberg, Richardson & Abraham, 2009*). Among the veteran population, CD is a moderately severe chronic disease that has modest resource consumption. However, the use of anti-TNF agents increases the overall direct costs associated with CD. Our results provide real world effectiveness of anti-TNF agents on CD in the veteran population; however, we did not evaluate whether the strategy was based on a top-down or step-up approach (*D'Haens, 2009*; *Hanauer, 2003*; *Lin, Blonski & Lichtenstein, 2010*). Debate continues on whether a top-down approach is more effective

and efficient relative to a step-up approach for CD treatment and management (*D'Haens, 2009*; *Hanauer, 2003*; *Lin, Blonski & Lichtenstein, 2010*).

We reported on anti-TNF agent use across a wide spectrum of different indications. We also presented the effectiveness of anti-TNF agents for the top three indications: RA, CD, and psoriasis, but small sample size prevented us from performing additional statistical tests. The high proportion of patients who were responders for RA, CD, and psoriasis provide some support for the effectiveness of anti-TNF agents at 12 months which parallels the results of other studies (*Breedveld et al., 2006*; *Colombel et al., 2010*; *Colombel et al., 2007*; *Kameda et al., 2010*; *Sandborn et al., 2007b*; *Weinblatt et al., 2003*; *Weinblatt et al., 1999*). Justification for using anti-TNF agents for these three indications will require a more robust analysis with a larger veteran population along with cost-effectiveness analyses.

Developing infection is a risk associated with using anti-TNF agents. Lane, et al. reported that VA patients using anti-TNF agents for RA from 1998 to 2005 were at risk of being hospitalized for an infection [Hazard Ratio (HR) = 1.24; 95% CI [1.02, 1.50]] (*Lane et al., 2011*). *Ford & Peyrin-Biroulet (2013)* reported that patients using anti-TNF agents for CD had higher risk of developing an opportunistic infection compared to placebo [Relative Risk (RR) = 2.05; 95% CI [1.10, 3.85]] (*Ford & Peyrin-Biroulet, 2013*). The risk of developing *Mycobacterium tuberculosis* was higher but not significant in patients receiving anti-TNF agents compared to placebo (RR = 2.52; 95% CI [0.62, 10.21]) (*Ford & Peyrin-Biroulet, 2013*). We reported that patients on etanercept and adalimumab developed infections; however, these did not require hospitalizations and were treated with oral antibiotics in the outpatient setting. Furthermore, two infection-related adverse events resulted in discontinuation of the anti-TNF agents. *Lane et al. (2011)* reported that patients receiving infliximab for RA had a higher hazard of hospitalized infections relative to etanercept (HR = 1.51; 95% CI [1.14, 2.00]); and patients receiving adalimumab had a lower but non-significant hazard of hospitalized infections relative to etanercept (HR = 0.95; 95% CI [0.68, 1.33]) (*Lane et al., 2011*). In our study, we reported that patients in the adalimumab group had more infections compared to the etanercept group; and no infections were reported in the infliximab group. This conflict may be due to the small sample size which potentially introduces type II error. Furthermore, *Lane et al. (2011)* focused on hospitalized infections in RA while our report described non-hospitalized infection events for all anti-TNF agent indications. In our study, stratifying by RA, we observed that 2 out of 7 patients receiving adalimumab developed an infection; however, infections were not observed in the other groups for RA (data not presented). Future studies will need to incorporate a larger sample size in order to capture any infection events stratified by disease.

Our study has limitations that are inherent to observational studies and studies involving chart reviews. This was a retrospective study that used manual chart reviews to abstract the relevant data. Consequently, there may be some validity issues with how responders and non-responders were determined. Published studies use standardized and validated criteria (ACR, DAS, and CDAI) to generate an objective score for a disease

(e.g., RA and CD). However, in practice, these criteria may not always be used or may be impractical. As a result, manual chart reviews are often necessary to determine response to therapy. Previous studies have demonstrated that manual chart reviews may be more sensitive in identifying cases of RA compared to using electronic medical record or ICD-9 coding (*Liao et al., 2010*; *Love, Cai & Karlson, 2011*; *Tinoco et al., 2011*). However, interpretation of the meaning and intention of the chart notes require careful attention to the signs and symptoms of disease and improvement in patient functionality. Misclassification may pose a potential source of internal validity; therefore, we took precautions and used two independent chart reviewers to mitigate this problem. This example highlights an important limitation with using chart review in determining response. Due to a lack of objective reporting, evaluation of success with anti-TNF agents would be reduced to evaluation based on a case definition of response. We acknowledge that misclassification is an important bias that cannot be truly ruled out. Ideally, an objective measurement should be recorded in the patient's chart; however, this has not been a requirement for reimbursement or continuation of anti-TNF agents. Future policy development may consider this as a need in order to accurately report response in patients receiving these costly agents.

We focused on a single site, which may not be generalizable to other VA institutions. Although each VA medical center abides by the VHA National Formulary, differences in practice may exist at individual sites. A lack of a VA national criteria or guideline for anti-TNF agents in RA and CD has led some sites to develop their own local criteria for use. These criteria may differ resulting in a variety of methods for providers to get access to anti-TNF agents for prescribing. In addition, our study focused on a single VA medical center population which limits generalizability to the general veteran population. Future studies will need to incorporate the entire VA population using anti-TNF agents to confirm our findings.

This study had missing data, which is a concern, especially if the missing data is informative. We chose to assume that the missing data was not informative. This does not rule out the possibility that bias exists. Caution should be applied when extrapolating what potential effect these missing data would have on the overall conclusion of this observational study.

Patients who were categorized as non-responders could have been switched to another anti-TNF agent, continued on the anti-TNF agent, or discontinued altogether. It was not possible to establish the average time that these patients were on an anti-TNF agent due to these issues. We reported that the average time to follow up was 86 days, which may not reflect the average follow up in the community. Further observational studies should evaluate the average time to follow up with anti-TNF agents in order to establish the optimal time to measure efficacy and safety.

We reported that several patients were on DMARDs at baseline. However, due to the small sample size, we were unable to evaluate whether they were meaningful differences with this population in terms of effectiveness and safety. Future studies should investigate

this population and whether increased effectiveness or worsening side effect profile is reported.

Finally, patients at the VA may have dual care with non-VA medical centers and providers. These patients may have experienced changes in their therapy and received treatment for infections that were not captured with the VA electronic records. Clinical trials have reported the proportion of patients with infections ranging from 5.7% (*Emery et al., 2009*) to 46% (*Colombel et al., 2010*). To complicate matters, patient healthcare benefits may not be restricted to the VA resulting in patients "shopping" for different providers. This may lead to vital information about the patient's disease and status that are not shared with the VA (*Nayar et al., 2013a*; *Nayar et al., 2013b*; *Weeks, Yano & Rubenstein, 2002*). As a result, there may be some underreporting of infection events with our analysis.

We did not observe golimumab and certolizumab pegol utilization at VASDHS between 2010 and 2011, despite their availability. We speculate that this was due to their novelty, lack of provider experience, and availability of alternative biologic agents (e.g., IL-6 inhibitors and integrin inhibitors). Although these other anti-TNF agents were not used at VASDHS, it is possible that they may have been utilized at different VA facilities. Future studies will need to expand this investigation to include more VA facilities in order to capture golimumab and certolizumab pegol utilization.

## CONCLUSION

A majority of patients who were initiated with an anti-TNF agent in the VA were categorized as responders at 12 months follow-up. This was observed for RA and CD indications. Infections were only observed in etanercept and adalimumab patients; however, low sample size in the infliximab subgroup may introduce type II error. Future studies will need to investigate the entire VA population using anti-TNF agents to determine if response is consistent with those reported at VASDHS.

### Funding
Dr. Bounthavong has received a grant from UCB pharmaceuticals, which is the manufacturer of Cimzia (certolizumab pegol). IIS#: 002296. The funders had no role in study design, data collection and analysis, decision to publish, or preparation of the manuscript.

### Grant Disclosures
The following grant information was disclosed by the authors:
IIS#: 002296.

### Competing Interests
Dr. Bounthavong has received a grant from UCB pharmaceuticals, which is the manufacturer of Cimzia (certolizumab pegol). Drs. Madkour and Kazerooni declare that there are no conflicts of interest regarding the publication of this article. The views and opinions of the authors do not reflect those of the US Department of Veterans Affairs.

## Author Contributions

- Mark Bounthavong conceived and designed the experiments, performed the experiments, analyzed the data, contributed materials/analysis tools, wrote the paper, prepared figures and/or tables, reviewed drafts of the paper, design, chart review, analysis, and writing.
- Nermeen Madkour contributed materials/analysis tools, wrote the paper, reviewed drafts of the paper, design, chart review, and writing.
- Rashid Kazerooni contributed materials/analysis tools, wrote the paper, reviewed drafts of the paper, writing.

## Ethics

The following information was supplied relating to ethical approvals (i.e., approving body and any reference numbers):

UCSD/Veterans Affairs San Diego Healthcare System Research and Development Institutional Review Board Protocol #: H120150.

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
