# Peer review of "Retrospective cohort study of anti-tumor necrosis factor agent use in a veteran population"

_PeerJ, doi:10.7717/peerj.385_

## Round 0.1 · original submission · Minor Revisions

The comments from reviewers have been favorable. Small sample size and retrospective design are limitations. Also, the findings can only be applied to a specific VA population, which composed largely of white males. However, the paper did report some important issues on the use of anti-TNF agents. Further larger studies will be needed.

Reviewer 1 ·

Basic reporting

Line 95-97 (last sentence): Please clarify if this reference point giving average annual cost per patient in the US for Crohns disease. Did this reference (Bodger 2002) not include the use of anti-TNF agents being used as a part of the disease management during the study period in order for you to show the difference in use of biologics to expenditures? (A study by Kappelman et al Gastroenterology 2008;1907-1913 reports a different number and proportion, but they did include anti-TNFs in the study period.)

Experimental design

Line 129. Did the patients have to be on the anti-TNF agent for a specified period of time in order to be included in the study? If not, please provide a brief summation of how long anti-TNF therapy is usually given for efficacy to be determined in the background section.

Line 203: What was the average duration of anti-TNF therapy for patients in the non-responder group?

Line 204: Please clarify, when you say 15 cases had infections that did not require hospitalization, are those 15 unique patients? Or were any of the cases, repeated by a patient with 2 or 3 cases/episodes? Is this infection rate similar to those reported in the literature with anti-TNF use?

Validity of the findings

Lines 215-217: Regarding the missing data for the 12 patients, was the data deemed clinically insignificant data to determine the response vs non-response outcome or confirm adverse effect data? If so, please indicate as such. Otherwise, please clarify how you categorized those patients into your analysis? Were they then put in the non-responder group?

Additional comments

It was a very well written paper with many robust statistical analysis performed to support the results and discussion.

·

Basic reporting

The manuscript by Bounthavong et al. provided a nice overview of the use of anti-TNF agents in the VA setting.

The manuscript provided an adequate introduction and background and demonstrates that the material presented is unique and novel.

Experimental design

The method section is adequately explained with sufficient detail.

The definitions for clinical efficacy seem to be vague.
The difference between responders and partial responders seem to be subjective.

The use of concomitant immunomodulators was mentioned but not included in the secondary aims.

Patients on comcomitant immunomodulators for example infliximab + azathioprine will have different outcomes than individuals on monotherapy with anti-TNF agents.

No mention was made regarding the severity of disease for example in Crohn's disease patients with mild to moderate disease will respond differently than patients with moderate to severe disease.

Validity of the findings

The overall design and primary aims are clearly stated.

The results of the study are limited by the sample size. It is difficult to generalize the findings of the study based on the the findings of 92 patients and even smaller sub groups (27 Crohn's disease and 24 RA).

Additional comments

Overall it is a nice study that attempts to provide some important information in the VA setting.

The methods and results can be made more robust by collecting more data. Consider collecting more than one year of data. Looking at concomitant use of immunomodulators.

---

## Round 0.2 · accepted · Accept

Authors have adequately addressed the concerns raised by reviewers.